# Single transcript level atlas of oxytocin and the oxytocin receptor in the mouse brain

Vitaly Ryu[1,2]*, Anisa Azatovna Gumerova[1,2], Georgii Pevnev[1,2], Funda Korkmaz[1,2], Hasni Kannangara[1,2], Liam Cullen[1,2], Farhath Sultana[1,2], Ronit Witztum[1,2], Steven Lee Sims[1,2], Tal Frolinger[1,2], Ofer Moldavski[1,2], Orly Barak[1,2], Emily Weiss[1,2], Jay J Cao[3], Daria Lizneva[1,2], Ki A Goosens[1,2], Tony Yuen[1,2], Mone Zaidi[1,2]*

[1]Institute for Translational Medicine and Pharmacology (ITMaP), Icahn School of Medicine at Mount Sinai, New York, United States; [2]Departments of Medicine and of Pharmacological Sciences, Icahn School of Medicine at Mount Sinai, New York, United States; [3]United States Department of Agriculture, Grand Forks Human Nutrition Research Center, Grand Forks, ND 58203, Grand Forks, United States

**Abstract** Oxytocin (OXT), a primitive nonapeptide known to regulate reproduction and social behaviors, is synthesized primarily in the hypothalamus and is secreted via the hypophyseal-portal system of the posterior pituitary gland. In line with the premise that pituitary hormones, traditionally thought of as regulators of single targets, display an array of central and peripheral actions, we found that OXT directly affects bone and body composition. The effect of OXT on bone remodeling is physiologically relevant, as elevated OXT levels during pregnancy and lactation cause calcium mobilization from the maternal skeleton for intergenerational calcium transfer towards fetal bone mineralization. There is an equally large body of evidence that has established the presence of OXT receptors (OXTRs) in the brain through which central functions, such as social bonding, and peripheral functions, such as the regulation of body composition, are exerted. To purposefully address effects of OXT on the brain, we used RNAscope to map OXT and OXTR expression, at the single transcript level, in the whole female and male mouse brains. Identification of brain nuclei with the highest OXT and OXTR transcript density sheds further light on functional OXT nodes that could be further interrogated experimentally to define new physiologic circuitry.

*For correspondence:
vitaly.ryu@mssm.edu (VR);
mone.zaidi@mountsinai.org (MZ)

## Editor's evaluation

This study provides a valuable, transcript-level map of OXT neurons and OXTR expression across the mammalian brain using advanced single-molecule RNAscope. The authors present compelling evidence supporting their conclusions, combining chromogenic assays with high-quality, state-of-the-art microscopy. By clearly delineating oxt and oxtr expression across multiple nuclei and brain regions relevant to behavior and physiology, the work substantially advances understanding of central oxytocin signaling and will be of broad interest to neuroscientists and endocrinologists.

## Introduction

Oxytocin (OXT), a neuropeptide synthesized primarily by magnocellular neurons within the paraventricular (PVH) and supraoptic nuclei (SON) of the hypothalamus (*Sofroniew, 1983*; *Swanson and Sawchenko, 1983*; *Landgraf and Neumann, 2004*), has been broadly implicated in the control of parturition, lactation, appetite, emotions, stress responses, and social behavior. The distribution of

OXT receptors (OXTRs) across the brain in different species provides a proxy for the distribution of OXT binding, thus providing evidence for OXT nodes in the brain of physiologic relevance. It has been reported that OXTRs are expressed in many brain sites, including the central nucleus of the amygdala and the ventromedial hypothalamic nucleus (VMH) (*Bale and Dorsa, 1995b*; *Bale and Dorsa, 1995a*). Furthermore, *Oxtr* mRNA has been detected in the hypothalamus, olfactory bulb, ventral pallidum, and the dorsal vagal nucleus (*Yoshimura et al., 1993*; *Adan et al., 1995*).

OXT mediates a variety of peripheral and central functions. While the peripheral actions comprise milk ejections, uterine contractions, and prolactin production, the central actions of OXT are mostly related to female reproduction, including sexual receptivity (*Caldwell et al., 1986*), pair bonding (*Insel, 1992*), and maternal behavior (*Fahrbach et al., 1984*; *Pedersen et al., 1982*; *Insel, 1990*). Central functions of OXT also include modulation of cardiac vagal input (*Bohus et al., 1996*), memory consolidation (*Dyball and Paterson, 1983*), and social/affiliative behavior (*Insel, 1992*; *van Wimersma Greidanus and Maigret, 1996*). Axons and dendrites of OXT neurons are localized in close proximity to the third ventricle and even in between tanycytes and ependymal cells facing the cerebrospinal fluid (*Landgraf and Neumann, 2004*). Notably, magnocellular OXT neurons send extended dendritic trees, forming the basis for the somato-dendritic release of OXT within the PVH and SON (*Ludwig and Leng, 2006*; *Neumann et al., 1993*; *Neumann, 2007*; *Pow and Morris, 1989*). Such release is likely to facilitate autocrine and/or paracrine regulation of OXT neurons towards physiologic demands, such as lactation (*Moos and Richard, 1989*; *Neumann et al., 1994*) and child birth (*Neumann et al., 1996*). To exert neuronal effects, locally released OXT binds to local OXTRs, which are expressed within or are juxtaposed to the target region, for example, on synapses, as well as on axons and glial processes (*Mitre et al., 2016*). Alternatively, OXT could putatively diffuse over longer distances to bind to adjacent OXTRs (*Landgraf and Neumann, 2004*; *Ludwig and Leng, 2006*; *Mitre et al., 2016*). Given that OXT exerts its multiple behavioral effects through its action on several regions of the forebrain and mesolimbic brain, the question of whether other extrahypothalamic projections of OXT neurons may also have a role garners significant importance.

It is also becoming increasingly clear that both anterior and posterior pituitary hormones, traditionally thought of as regulators of single physiological processes, affect multiple bodily systems, either directly or via actions on brain receptors (*Zaidi et al., 2018*; *Abe et al., 2003*). Nontraditional actions of OXT include its ability to affect the skeleton, wherein it stimulates bone formation by osteoblasts and modulates the function of bone-resorbing osteoclasts (*Sun et al., 2019*).

Despite a *corpus* of evidence for the expression of OXT and OXTRs in various brain regions, and their function in regulating central and peripheral actions, such as social behavior and satiety (*Sun et al., 2019*; *Bale et al., 2001*), there remains the need for a detailed, sex-specific mapping of the anatomical geography of the OXT and OXTR systems in the brain. Here, we use RNAscope—a cutting-edge technology that detects single RNA transcripts—to create a comprehensive sex-specific atlas of the OXT and OXTR in the mouse brain. We believe that this compendium of OXT and its receptor in concrete brain sites should provide a resource for investigators to study both peripheral and central effects of interrogating OXTRs site—specifically in health and disease. Our identification of brain nuclei with the highest OXT and OXTR transcript density will thus deepen our future understanding of the functional engagement of the central OXT-containing neuronal nodes within a large-scale functional network.

## Results

Mapping autoradiographic studies suggest that the distribution of OXTRs in the brain varies greatly among different rodent species (*Dubois-Dauphin et al., 1992*; *Elands et al., 1988*; *Insel et al., 1997*; *Tribollet et al., 1992*). Besides mapping the full anatomical distribution of *Oxt* and *Oxtr* by RNAscope, the present study also assessed sex differences in *Oxt* and *Oxtr* distribution. Allowing the detection of single transcripts, RNAscope uses ~20 pairs of transcript-specific double *Z*-probes to hybridize 10-μm-thick whole-brain sections. Preamplifiers first hybridize to the ~28-bp binding site formed by each double *Z*-probe; amplifiers then bind to the multiple binding sites on each preamplifier; and finally, labeled probes containing a fluorescent molecule bind to multiple sites of each amplifier.

RNAscope data were quantified on sections from coded three female and three male mice. Each section was viewed and analyzed using CaseViewer 2.4 (3DHISTECH, Budapest, Hungary) and QuPath v.0.2.3 (University of Edinburgh, UK). The *Atlas for the Mouse Brain in Stereotaxic Coordinates*

(*Paxinos and Franklin, 2007*) was used to identify every nucleus, sub-nucleus, or region, which was followed by manual counting of *Oxt* and *Oxtr* transcripts by two independent observers (VR and AG) in every tenth section using a tag feature. Receptor density was calculated by dividing the transcript number by the area ($\mu m^2$, ImageJ) in every nucleus, sub-nucleus, or region. Photomicrographs were prepared using Photoshop CS5 (Adobe Systems) only to adjust brightness, contrast, and sharpness, and to remove artifacts (i.e., obscuring bubbles).

In males, we report the expression of the *Oxtr* in 359 mouse brain nuclei, sub-nuclei, and regions. Probe specificity was established by a positive signal in the epididymis with an absent signal in the liver (negative control) (*Figure 1A*). Notably, *Oxtr* transcripts were detected bilaterally, with no apparent ipsilateral domination. Transcript density was highest in ventricular regions, followed, in descending order, by the hypothalamus, olfactory bulb, hippocampus, cerebral cortex, medulla, midbrain and pons, forebrain, thalamus, and cerebellum (*Figure 1B*). Using the RNAscope dataset, we further calculated *Oxtr* density in various brain nuclei, sub-nuclei, and regions. High *Oxtr* transcript densities and counts, respectively, were also noted in several nuclei, sub-nuclei, and regions as follows (*Figure 1C*): ventricular regions—ependyma of the OV and 3V; hypothalamus—AHiPM for both; olfactory bulb—vn and GrO; hippocampus—Py for both; cerebral cortex—Cl and Pir; medulla—10N and Sp5I; midbrain and pons—IPF and DpMe; forebrain—aci and CPu; thalamus—PV and PVA and cerebellum—Sim for both (see Appendix 1 for nomenclature and *Figure 1—figure supplement 1* for transcript count and representative photomicrographs).

In females, we report the expression of the *Oxtr* in 301 mouse brain nuclei, sub-nuclei, and regions. Probe specificity was again established by a positive signal in the ovary with no signal in the liver (*Figure 2A*). Transcript density was highest in the hippocampus, followed, in descending order, by the olfactory bulb, hypothalamus, cerebral cortex, ventricular regions, forebrain, medulla, thalamus, midbrain and pons, and cerebellum (*Figure 2B*). High *Oxtr* transcript densities and counts, respectively, were also noted in several nuclei, sub-nuclei, and regions as follows (*Figure 2B*): hippocampus—Py for both; olfactory bulb—AOD and GrO; hypothalamus—SO and PMCo; cerebral cortex—AIP and Pir; ventricular regions—ependyma of the OV and SVZ; forebrain—SFO and aci; medulla—10N for both; thalamus—PV for both; midbrain and pons—EW and PAG and cerebellum—6Cb for both (see Appendix 1 for nomenclature and *Figure 2—figure supplement 1* for transcript count and representative photomicrographs).

RNAscope also revealed *Oxt* expression in the hypothalamus and forebrain of both male and female mice (*Figure 3A and B*). High *Oxt* counts were detected in several nuclei, sub-nuclei, and regions of females and males, respectively, as follows (*Figure 3C*): hypothalamus—PaMP and PaMM and forebrain—MPA and LPO. Overall, the numbers of *Oxt*-expressing cells were markedly greater in female compared to the male mice. That is, we found 22 hypothalamic and forebrain regions with 486 *Oxt*-positive cells in the female mouse. In contrast, there were 15 hypothalamic and forebrain regions containing 308 *Oxt*-positive cells in the male brain. Breaking this down, in the hypothalamus, the number of *Oxt*-positive cells was 372 in the female compared with 228 *Oxt*-positive cells in the male. The number of *Oxt*-positive cells was 114 in the female forebrain compared with 80 *Oxt*-positive cells in the male forebrain.

*Oxtr* expression was also mapped in regions and sub-regions within the hypothalamus (*Figure 1* and *Figure 1—figure supplement 1*). Certain of these hypothalamic sub-regions, such as the lateral hypothalamus (LH) and dorsomedial hypothalamus (DM), send sympathetic nervous system (SNS) outflow to both bone and fat tissue (*Ryu et al., 2015*; *Ryu et al., 2017*). Additionally, RNAscope also showed *Oxtr* expression in both anterior and posterior pituitary lobes (*Figure 4A*), with *Oxtr* transcript density that was markedly higher in the female compared with male mice (*Figure 4B*).

We also found that six hypothalamic nuclei, sub-nuclei, and regions in male mice displayed overlapping *Oxt* and *Oxtr* transcripts. *Oxtr/Oxt* ratios within the same brain site were as follows: 0.70 in the medial parvicellular part of the paraventricular hypothalamic nucleus (PaMP); 1.51 in the medial magnocellular part of the paraventricular hypothalamic nucleus (PaMM); 6.39 in the lateral magnocellular part of the paraventricular hypothalamic nucleus (PaLM); 26.8 in the arcuate nucleus (Arc); 109.50 in the medial amygdaloid nucleus (MeA); 151.13 in the tuber cinereum area (TC), and 222.00 in the lateroanterior hypothalamic nucleus (LA). In contrast, we found three forebrain nuclei, sub-nuclei, and regions in female mice with overlapping *Oxt* and *Oxtr* transcripts. *Oxtr/Oxt* ratios within the same brain site were as follows: 1.35 in the anterior commissural nucleus (AC); 3.50 in the medial preoptic

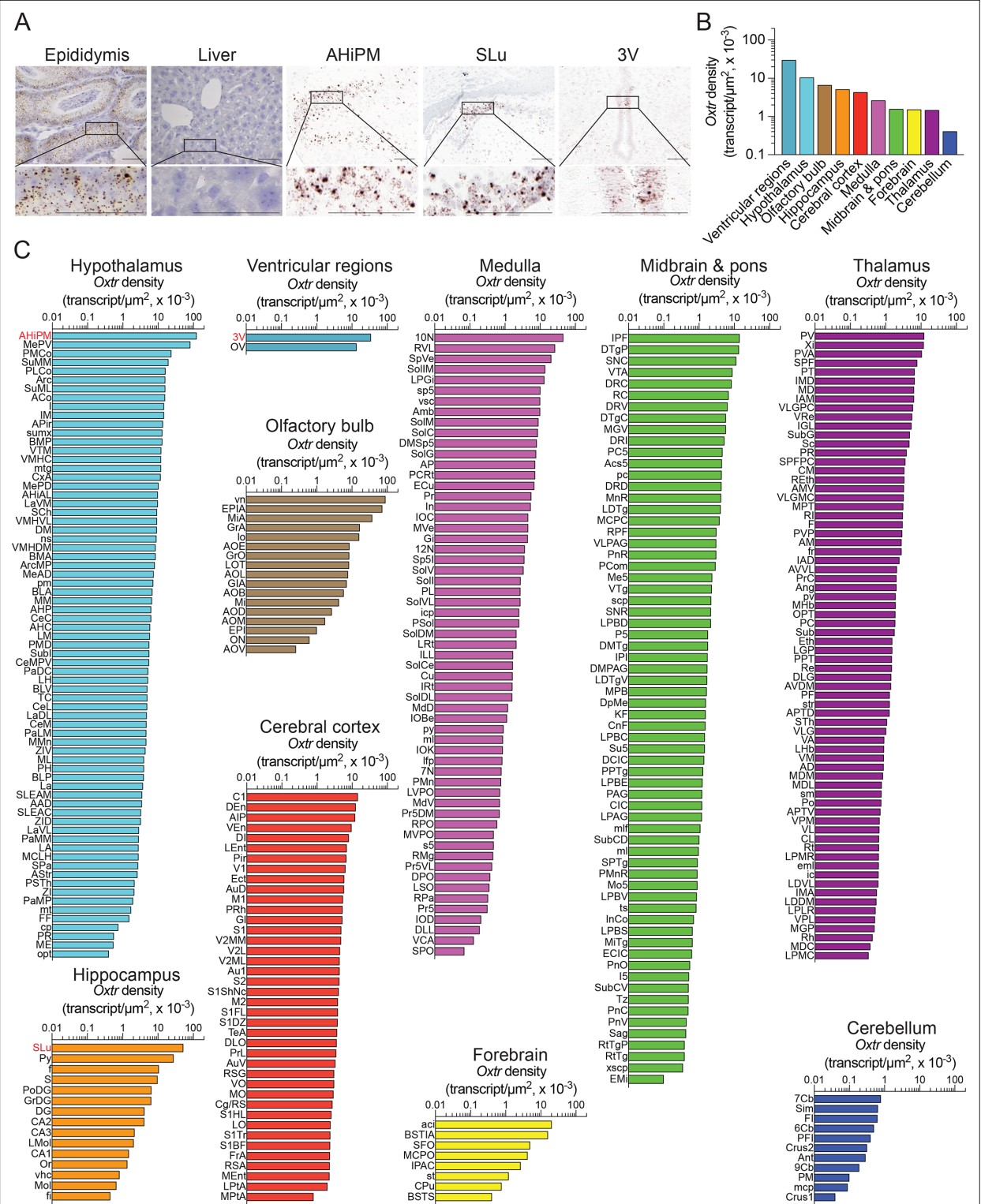

**Figure 1.** *Oxtr* expression in the male brain. (**A**) RNAscope revealed *Oxtr*-positive transcripts in the epididymis, but not in the liver (positive and negative controls, respectively). Also shown are representative micrographs of the posteromedial part of the amygdalohippocampal area (AHiPM) of the hypothalamus, pyramidal cell layer (Py) of the hippocampus, and the third ventricle (3V) of the ventricular regions. Scale bar: 50 μm. (**B**) *Oxtr* transcript density in the brain regions detected by RNAscope. (**C**) *Oxtr* transcript density in nuclei, sub-nuclei, and regions of the ventricular system, hypothalamus, olfactory bulb, hippocampus, cerebral cortex, medulla, midbrain and pons, forebrain, thalamus, and cerebellum.

The online version of this article includes the following source data and figure supplement(s) for figure 1:

*Figure 1 continued on next page*

*Figure 1 continued*

**Source data 1.** *Oxtr* densities in brain nuclei, subnuclei and regions.

**Figure supplement 1.** *Oxtr* expression in the male brain.

**Figure supplement 1—source data 1.** *Oxtr* transcript numbers in brain nuclei, subnuclei and regions.

nucleus, medial part (MPOM), and 13.0 in the bed nucleus of the stria terminalis, medial division, and posterolateral part (BSTMPL). As with the hypothalamus, *Oxtr/Oxt* ratios in ten nuclei, sub-nuclei, and regions were 0.05 in the PaLM; 0.18 in the PaMM; 0.82 in the paraventricular hypothalamic nucleus, anterior parvicellular part (PaAP); 1.74 in the LA; 5.83 in the anterior hypothalamic area, posterior part (AHP); 8.00 in the anterior hypothalamic area, anterior part (AHA); 9.33 in the supraoptic nucleus (SO); 39.0 in the ventromedial hypothalamic nucleus, dorsomedial part (VMHDM); 40.7 in the TC and 75.3 in the lateral hypothalamic area (LH).

## Discussion

Here, we supplement and integrate previous information on OXT and OXTR expression in the murine brain and report, for the first time, abundant OXTR expression in 301 and 359 brain nuclei, sub-nuclei, and regions in females and males, respectively, as well as, importantly, sex-specific *Oxt* and *Oxtr* expression. This report is thus the most comprehensive atlas of brain *Oxt* and *Oxtr* expression at the single transcript level. Expression of both *Oxt* and *Oxtr*, particularly in overlapping hypothalamic sub-nuclei, nuclei, and regions, points to functionally active neuronal nodes within a large-scale OXT-OXTR network in the brain.

It has been reported that cell bodies and dendrites of OXT-producing neurons within the PVH and SON release OXT and AVP within the magnocellular nuclei, where we find the highest *Oxtr/Oxt* colocalization—this suggests an additional, possibly paracrine action of OXT (*Ludwig and Leng, 2006*; *Neumann et al., 1993*; *Neumann, 2007*; *Pow and Morris, 1989*). Indeed, locally released OXT is involved in pre- and post-synaptic modulation of the electrical activity (*Bourque et al., 1993*; *Shibuya et al., 2000*; *Kombian et al., 1997*). Similar to magnocellular neurons of the PVH, we find that several *Oxt*-producing neuronal populations also overlap with *Oxtr* expression in other hypothalamic sites— hereby termed '*Oxtr/Oxt* nodes'—these include Arc, MeA, TC, and LA.

It is now well known that central OXT decreases ingestive behavior while OXTR antagonism has the opposing effect in rodents (*Liu et al., 2021*; *Arletti et al., 1990*; *Blevins et al., 2016*; *Klockars et al., 2018*; *Liu et al., 2020*; *Noble et al., 2014*; *Ong et al., 2015*). Of note is that, in addition to the nucleus of the solitary tract (NTS) (*Ong et al., 2015*), the *Oxtr/Oxt* node in the Arc (and, possibly, Arc-bordering LA) rapidly induces satiety (*Fenselau et al., 2017*) and suppresses excessive food intake to control ingestive behavior (*Inada et al., 2022*). In terms of sex-specific ability of OXT to inhibit ingestive behavior, it has been reported that the capacity of OXT to decrease food intake is attenuated in females compared with males, whereas lower OXT doses are effective at reducing food intake in males, and doses that are effective in both sexes reduce consumption for a longer duration in males (*Liu et al., 2020*). The *Oxtr/Oxt* node in the MeA likely explains the paracrine regulation by OXT of male preference for females and their scents (*Yao et al., 2017*). Thus, the ablation of *Oxtr* in aromatase-expressing neurons of the MeA fully recapitulates the elimination of female preference in males, suggesting that this node is both necessary and sufficient for social recognition (*Ferguson et al., 2001*). Lastly, the TC is a sheet of gray matter that forms a median eminence (ME) around the base of the pituitary stalk or infundibulum; therefore, the *Oxtr/Oxt* node in the TC (and tanycyte) could be important for mediating bidirectional brain–periphery crosstalk by modulating the blood–hypothalamus brain barrier.

Although we detected clear sex differences and similarities in *Oxtr* transcript expression in multiple brain areas, here we will focus on those associated with stress, energy homeostasis, emotional, and affective behaviors. Surprisingly, the highest *Oxtr* transcript density was noted in the ependymal layers of the OV and 3V in both sexes with greater expression density in males and, not surprisingly, in the hypothalamus (*Sofroniew, 1983*; *Swanson and Sawchenko, 1983*; *Landgraf and Neumann, 2004*). In the hypothalamus, the highest density was found in the posteriomedial part of the amygdalohippocampal area (AHiPM) of males compared to that in females. It has been reported that in male mice, ~40% of *Oxtr*-positive neurons of the amygdalohippocampal area (AHi) project to the

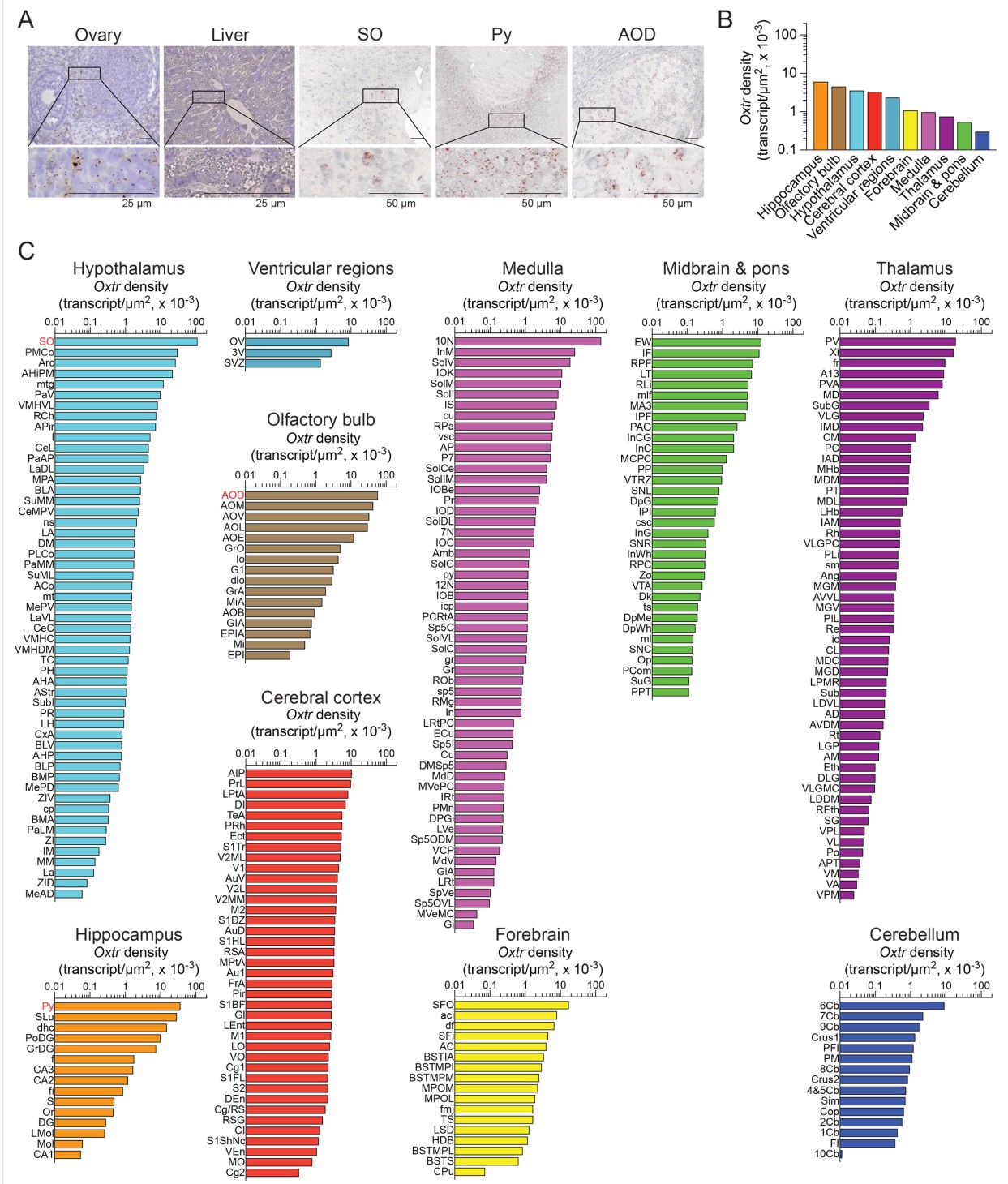

**Figure 2.** *Oxtr* expression in the female brain. (**A**) Representative micrographs of the supraoptic nucleus (SO) of the hypothalamus, the pyramidal cell layer (Py) of the hippocampus, and anterior olfactory nucleus, dorsal part (AOD) of the olfactory bulb are shown. Ovary and liver served as positive and negative controls, respectively. Scale bar: 50 μm. (**B**) *Oxtr* transcript density in the brain regions detected by RNAscope. (**C**) *Oxtr* transcript density in nuclei, sub-nuclei, and regions of the hippocampus, olfactory bulb, hypothalamus, cerebral cortex, ventricular system, forebrain, medulla, thalamus, midbrain and pons, and cerebellum.

The online version of this article includes the following source data and figure supplement(s) for figure 2:

**Source data 1.** *Oxtr* densities in brain nuclei, subnuclei and regions.

**Figure supplement 1.** *Oxtr* expression in the female brain.

*Figure 2 continued on next page*

*Figure 2 continued*

**Figure supplement 1—source data 1.** *Oxtr* trancript numbers in brain nuclei, subnuclei and regions.

medial preoptic area (MPOA) (***Sato et al., 2020***). Activation of these neurons, comprising excitatory projections to the MPOA, enhances exclusively an aggressive, but not parental behavior, towards pups (***Sato et al., 2020***). Of interest, females display the highest *Oxtr* density in the arcuate nucleus (Arc) compared to that of males. Arc$^{\text{Vglut2}}$ neurons have been reported to express the gene encoding *Oxtr* (***Fenselau et al., 2017***). Given that intra-Arc OXT acutely suppresses food intake and OXT exerts a direct stimulatory effect on Arc-OXTR neurons, it is plausible that the Arc-OXTR-satiety circuit, at least, responding to diet-induced hyperphagia (***Maric et al., 2022***), is pronounced in female rather

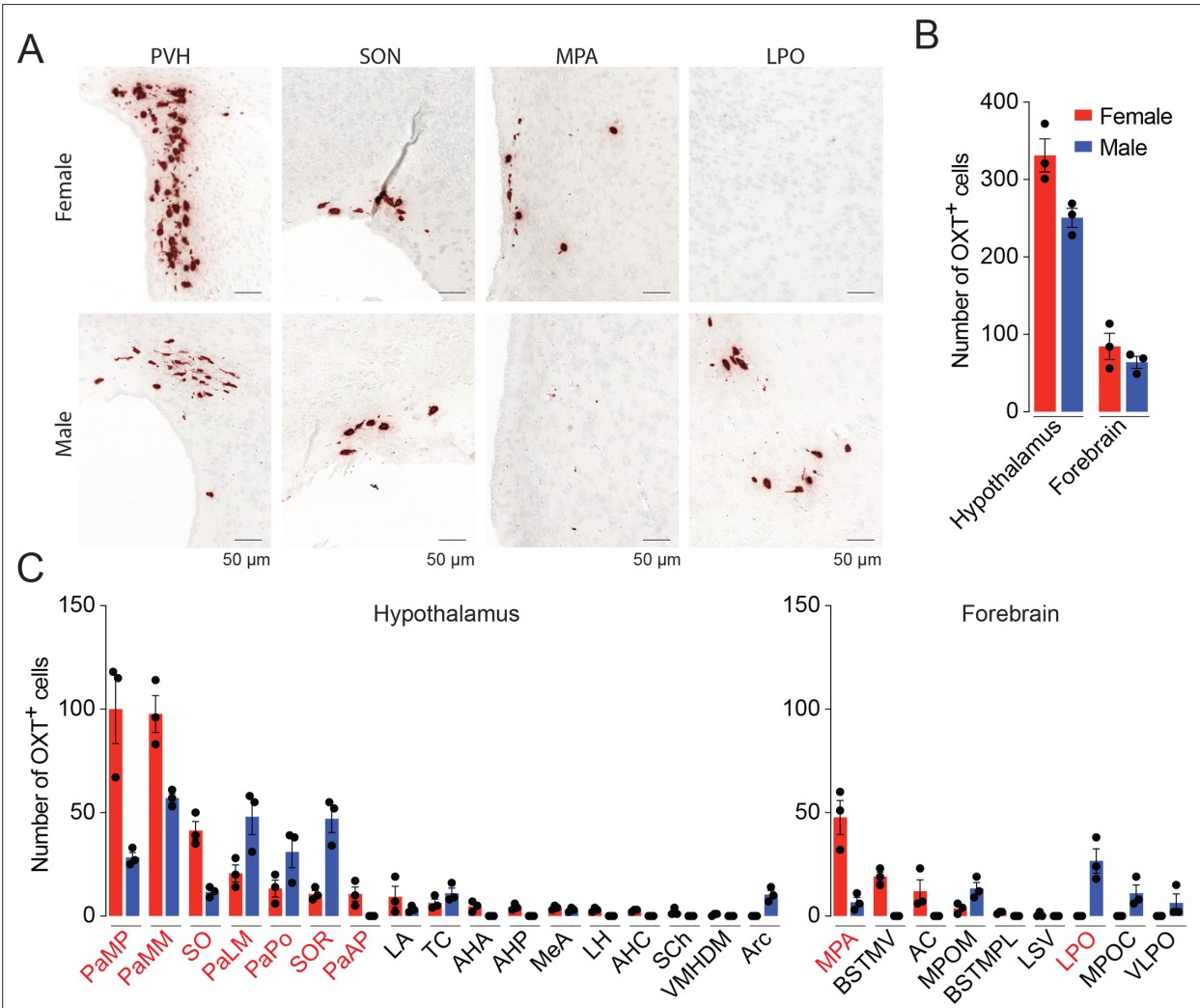

**Figure 3.** Sex differences in *Oxt* expression in the mouse brain. (**A**) The neurohypophysial hormone OXT is synthesized by magnocellular neurons primarily located in the PVH and SON hypothalamic nuclei. The magnocellular neurons send extended axonal projections into the neurohypophysis where OXT is released into the circulation in response to physiological demands. Therefore, PVH and SON served as positive controls for OXT expression in the brain. MPA: medial preoptic area; LPO: lateral preoptic area of the forebrain. Scale bar: 50 μm. (**B**) Sex differences in the numbers of *Oxt*-expressing neurons in nuclei, sub-nuclei, and regions of the hypothalamus and forebrain detected by RNAscope. (**C**) Total numbers of *Oxt*-expressing neurons in nuclei, sub-nuclei, and regions of the hypothalamus and forebrain of male and female mice. (**D**) *Oxt* transcript density in nuclei, sub-nuclei, and regions of the hypothalamus and forebrain of male and female mice. N = 3, values are shown as means ± SE. Student's *t*-test.

The online version of this article includes the following source data for figure 3:

**Source data 1.** Numbers of OXT-positive cells in the hypothalamus and forebrain.

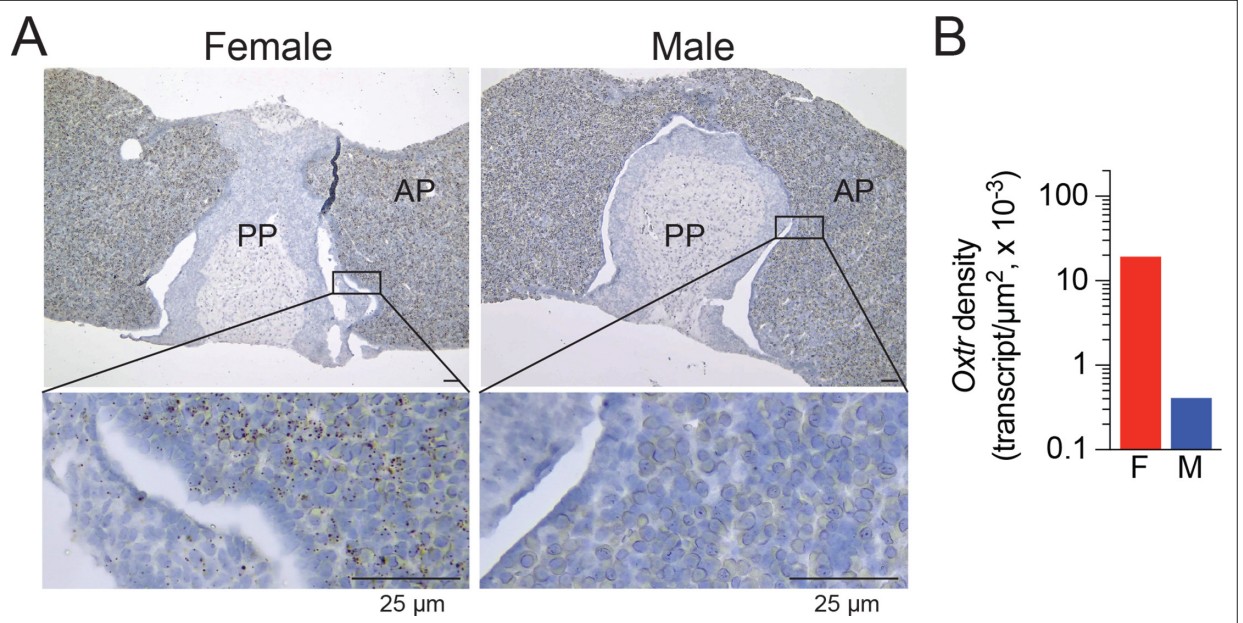

**Figure 4.** Sex differences in *Oxtr* expression in the pituitary gland. (**A**) Representative photomicrographs showing sex differences in *Oxtr* expression in anterior (AP) and posterior (PP) lobes of the pituitary gland detected by RNAscope. Scale bar: 25 μm. (**B**) Quantification of *Oxtr* transcript density in the pituitary gland of female and male mice (n=3).

The online version of this article includes the following source data for figure 4:

**Source data 1.** Sex-specific *Oxtr* densities in the pituitary gland.

than male mice. Indeed, it was demonstrated that female rats and mice display a lower than male level of diet-induced overeating (*Maric et al., 2022*).

The male olfactory bulb and hippocampus also displayed abundant *Oxtr* transcripts, with the highest density in the vomeronasal nerve (vn) and the pyramidal cell layer of the hippocampus (Py), respectively, in comparison to the female anterior olfactory nucleus, dorsal part (AOD) as well as the same (Py) hippocampal subregion. It has been reported that *Oxtrs* are expressed in the vomeronasal organ, an olfactory sensory structure involved in the detection of non-volatile chemosignals. OXT injection in mice has been shown to reduce pup-induced vomeronasal activity and aggressive behavior (*Nakahara et al., 2020*). Given vomeronasal activity declines as males grow up from a pup-aggressive state to a non-aggressive parental state, high *Oxtr* expression in the vn might indicate a functional switch from pup-aggressive behavior towards strengthening social and sexual behaviors during adolescence and adulthood. The highest *Oxtr* transcript densities in the AOD, AOM, and AOV of females are consistent with OXT function in the anterior olfactory region, particularly in relation to social cue processing and social recognition (*Oettl et al., 2016*). As with the hippocampal Py, *Oxtrs* are found in both excitatory and inhibitory pyramidal neurons within the CA2 and CA3 subregions of the hippocampus, suggesting that OXTRs in the Py may have a role in local circuits relating to stress, emotional, and affective behaviors.

In the cortex, we found that agranular insular cortex, posterior part (AIP) of females and claustrum (Cl) of males displayed the highest *Oxtr* transcript density. PVH- and SON-OXT neurons project to a wide range of cortical and limbic structures including AIP, Cl, hippocampus, medial amygdala, and the lateral septum, all of which comprise the social recognition memory circuit (*Mitre et al., 2016*; *Ferguson et al., 2001*; *Son et al., 2022*; *Tanimizu et al., 2017*; *Wang and Zhan, 2022*). SON neurons, upon activation by the OXTR, release OXT—a putative paracrine loop. Moreover, OXTRs mediate cardiac sympathetic stimulation through direct PVH projections to the intermediolateral column of the spinal cord (*Japundžić-Žigon, 2013*). Such reciprocal communications are supported by the studies inferring that affiliative social interactions increase OXT activity, which is followed by an anti-stress response, thus promoting bonding, relaxation, and growth, while reducing cardiovascular

and neuroendocrine stress (*Grippo et al., 2009*; *Krause et al., 2011*; *Lee et al., 2005*; *Windle et al., 1997*; *Wsol et al., 2008*).

Both males and females had the highest *Oxtr* transcript density in the medullary dorsal motor nucleus of vagus (10N), as has been shown previously in the rat (*Raggenbass et al., 1988*; *Dreifuss et al., 1988*; *Raggenbass et al., 1987*). The OXT-sensitive vagal neurons are mostly preganglionic motor neurons, projecting to the cervical, thoracic, and abdominal visceral areas (*Raggenbass et al., 1987*). It has also been shown that the microinjection of an OXT antagonist into the 10N blocks the increase in gastric acid secretion and bradycardia induced by electrical stimulation of the PVH—this suggests a role for central OXT in autonomic efferent activity (*Rogers and Hermann, 1986*).

Finally, we have recently published an atlas of pituitary glycoprotein hormone receptors, namely *Tshr*, *Fshr*, and *Lhcgr*, in more than 400 brain sites (*Ryu et al., 2022*). Surprisingly, we find a striking overlap in receptor distribution among the four receptors, including the *Oxtr*—with highest transcript levels in the ependymal layer of the third ventricle and olfactory bulb. While the role of olfactory OXTRs in social recognition is well established (*Oettl et al., 2016*; *Sun et al., 2021*; *Oettl and Kelsch, 2018*), the functional significance of OXTRs in the ependymal layer is yet unknown. However, in light of ubiquitous and newly emerging OXTR expression in the brain and peripheral organs, ependymal OXTRs seem to have an important role in gating the bidirectional brain–periphery crosstalk.

Despite higher plasma OXT levels in women than in men (*Marazziti et al., 2019*), prior, largely immunohistochemistry-based studies failed to identify a sex difference in *Oxt* expression in the brain. Similar numbers of OXT-positive immunoreactive (-IR) neurons were found in the PVH, SON, MPOA, and bed nucleus of stria terminalis (BNST) of prairie, pine, meadow, and montane voles (*Wang et al., 1996*), PVH and SON of naked mole rats (*Rosen et al., 2008*), and PVH, MPOA, LH, and anterior hypothalamus (AH) of long-tailed hamsters (*Xu et al., 2010*). Furthermore, no sex differences were detected in OXT-IR neurons in the PVH, SON, BNST, MeA in several species of non-human primates (*Caffé et al., 1989*; *Wang et al., 1997a*; *Wang et al., 1997b*). There were also no sex differences in *Oxt* mRNA expression in the PVH and SON of the rat (for review, see *Dumais and Veenema, 2016*). Lastly, there were no sex differences in the number or size of OXT neurons in the PVH and SON in humans (*Wierda et al., 1991*; *Fliers et al., 1985*; *Ishunina and Swaab, 1999*). By contrast, here we establish sex differences in *Oxt* expression in the mouse brain. Both the hypothalamus and forebrain of the females contained visibly more *Oxt*-positive cells compared with males. Whereas as expected, hypothalamic PVH of both sexes had high *Oxt* expression, the medial preoptic area (MPA) of the female forebrain and the lateral preoptic area (LPO) of the male forebrain contained the highest number of *Oxt*-expressing cells.

The neuroanatomical reality of the brain–bone–fat SNS feedback loops suggests coordinated and/or multiple redundant control of bone and fat remodeling (*Ryu et al., 2024*). We have noted that regions, such as the LH, DM, tuber cenereum area (TC), basolateral amygdaloid nucleus, and others, known to send SNS outflow to both bone and adipose tissues, express the *Oxtr* (*Ryu et al., 2015*; *Ryu et al., 2017*). Surprisingly, female mice had visibly fewer *Oxtr* counts in aforementioned sites compared to males, perhaps due to the organizational and activational effects of sex hormones (*Kammel and Correa, 2020*). This raises the possibility that certain actions of OXT on peripheral tissues, such as on body composition and bone, may also be mediated centrally. Indeed, non-classical actions of OXT include its ability to affect bone remodeling, wherein it stimulates bone formation by osteoblasts and modulates the function of bone-resorbing osteoclasts (*Sun et al., 2019*). We have also shown that OXT and vasopressin have opposing skeletal actions—effects that may relate to the pathogenesis of bone loss in pregnancy and lactation, and in chronic hypona-tremia, respectively (*Sun et al., 2019*; *Sun et al., 2016*; *Tamma et al., 2009*; *Tamma et al., 2013*). As with fat remodeling, it has been demonstrated that mice deficient in either OXT or OXTRs develop late-onset obesity despite normal ingestive behavior (*Takayanagi et al., 2008*). Moreover, the increased body weight in OXT knockout mice is accompanied by a 40% increase in abdominal adiposity (*Camerino, 2009*).

In all, studies on central OXT signaling and its control of reproductive, metabolic, and ingestive functions, and social behaviors occupy the vast majority of the literature. It is our hope that this comprehensive compendium of sex-specific *Oxt* and *Oxtr* expression in the brain will stimulate further investigations by others. In more general terms, the direct mapping of receptor expression in the brain and periphery provides the framework for determining new functions of ancient pituitary hormones

and helps refocus at least some in the field towards paradigm-shifting discoveries of non-traditional, multifaceted roles of OXT.

## Methods

### Mice

Adult C57BL/6J mice (~3–4-month-old) were housed in a 12 h:12 h light:dark cycle at 22 ± 2°C with ad libitum access to water and regular chow. All procedures were approved by the Mount Sinai Institutional Animal Care and Use Committee and are in accordance with Public Health Service and United States Department of Agriculture guidelines. Ethical approval for all experimental procedures wasobtained from the appropriate Institutional Review Board under protocol number PROTO202100038.

### RNAscope

Brains and pituitary glands were collected from male and female mice (n=3) for RNAscope. Briefly, mice were anesthetized with isoflurane (2–3% in oxygen; Baxter Healthcare, Deerfield, IL) and transcardiacally perfused with 0.9% heparinized saline followed by 10% neutral buffered formalin (NBF). Brains were extracted, sectioned into 0.5 cm (whole pituitary and adrenal glands) thick slices, and post-fixed in 10% NBF for 24 h, dehydrated, and embedded into paraffin. Coronal sections were cut at 5 µm with every tenth section mounted onto ~60 slides with three sections on each slide. This method allowed us to cover the entire brain and eliminate the likelihood of counting the same transcript twice. Sections were air-dried overnight at room temperature and stored at 4°C until required.

Detection of mouse *Oxt* and *Oxtr* was performed separately on paraffin sections using Advanced Cell Diagnostics (ACD) RNAscope 2.5 LS Multiplex Reagent Kit (#322100) and two RNAscope 2.5 LS probes, namely Mm-OXT (#493178) and Mm-OXTR (#412178). Epididymis/ovary and liver served as positive and negative controls for *Oxtr*, respectively. As with OXT, magnocellular cells of the PVH and SON served as positive controls while the brain from *Oxt* knockout mouse served as a negative control.

Slides were baked at 60°C for 1 h, deparaffinized, incubated with hydrogen peroxide for 10 min at room temperature, pretreated with Target Retrieval Reagent (#322001) for 20 min at 100°C and with Protease III for 30 min at 40°C. Probe hybridization and signal amplification were performed as per the manufacturer's instructions for chromogenic assays.

Following the RNAscope assay, the slides were scanned at ×20 magnification, and the digital image analysis was successfully validated using the CaseViewer 2.4 (3DHISTECH, Budapest, Hungary) software. The same software was employed to capture and prepare images for the figures in the article. Images of control tissues were taken using a Leica DM 1000 microscope. Detection of *Oxtr-* and *Oxt-* positive cells was also performed using the QuPath-0.2.3 (University of Edinburgh, UK) software based on receptor intensity thresholds, size, and shape.

### Statistical analysis

Data were analyzed by two-tailed Student's *t*-test and one-way repeated measures analysis of variance followed by Holm–Sidak's or Bonferroni's least significant difference post hoc tests using GraphPad Prism 10 (Boston, MA). Significance was set at $p < 0.05$. For simplicity and clarity, exact test results and exact p values are not presented.

## Acknowledgements

Work at the Icahn School of Medicine at Mount Sinai carried out at the Institute for Translational Medicine and Pharmacology was supported by R01 AG071870, R01 AG074092, and U01 AG073148 to TY and MZ; and U19 AG060917 and R01 DK113627 to MZ. JJC contributed to the concept and discussion of the study and was supported by the Agricultural Research Service of the United States Department of Agriculture, #3062-51000-053-00D. Mention of trade names or commercial products in this publication is solely for the purpose of providing specific information and does not imply recommendation or endorsement by the US Department of Agriculture. The findings and conclusions in this manuscript are those of the authors and should not be construed to represent any official USDA or US Government determination or policy.

# Additional information

## Competing interests

Vitaly Ryu, Jay J Cao, Daria Lizneva, Ki A Goosens: Reviewing editor, eLife. Tony Yuen: Senior editor, *eLife*. Mone Zaidi: consults for Gershon Lehmann, Guidepoint and Coleman groups. The other authors declare that no competing interests exist.

## Funding

| Funder | Grant reference number | Author |
| --- | --- | --- |
| National Institute on Aging | R01 AG071870 | Tony Yuen<br>Mone Zaidi |
| National Institute on Aging | R01 AG074092 | Tony Yuen<br>Mone Zaidi |
| National Institute on Aging | U01 AG073148 | Tony Yuen<br>Mone Zaidi |
| National Institute on Aging | U19 AG060917 | Mone Zaidi |
| National Institute of Diabetes and Digestive and Kidney Diseases | R01 DK113627 | Mone Zaidi |
| Agricultural Research Service of the United States Department of Agriculture | #3062-51000-053-00D | Jay J Cao |

The funders had no role in study design, data collection and interpretation, or the decision to submit the work for publication.

## Author contributions

Vitaly Ryu, Formal analysis, Investigation, Methodology, Writing – original draft; Anisa Azatovna Gumerova, Investigation, Visualization, Methodology; Georgii Pevnev, Liam Cullen, Ronit Witztum, Steven Lee Sims, Tal Frolinger, Ofer Moldavski, Investigation; Funda Korkmaz, Validation, Investigation; Hasni Kannangara, Formal analysis, Validation; Farhath Sultana, Emily Weiss, Formal analysis, Investigation; Orly Barak, Daria Lizneva, Investigation, Project administration; Jay J Cao, Resources, Investigation; Ki A Goosens, Data curation, Supervision; Tony Yuen, Conceptualization, Supervision, Funding acquisition, Project administration, Writing – review and editing; Mone Zaidi, Conceptualization, Funding acquisition, Writing – original draft, Project administration, Writing – review and editing

## Author ORCIDs

Vitaly Ryu https://orcid.org/0000-0001-8068-4577
Georgii Pevnev https://orcid.org/0000-0003-2015-9310
Funda Korkmaz https://orcid.org/0000-0002-9174-8369
Steven Lee Sims https://orcid.org/0000-0002-1636-084X
Ki A Goosens https://orcid.org/0000-0002-5246-2261
Mone Zaidi https://orcid.org/0000-0001-5911-9522

## Ethics

All procedures were approved by the Mount Sinai Institutional Animal Care and Use Committee and are in accordance with Public Health Service and United States Department of Agriculture guidelines.

## Decision letter and Author response

Decision letter https://doi.org/10.7554/eLife.95215.sa1
Author response https://doi.org/10.7554/eLife.95215.sa2

# Additional files

## Supplementary files

MDAR checklist

## Data availability

All data generated or analyzed during this study are included in the manuscript and supporting files; source data files have been provided for *Figures 1–4*, *Figure 1—figure supplement 1* and *Figure 2—figure supplement 1*.

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

# Appendix 1

**Appendix 1—table 1.** Glossary of the brain nuclei, sub-nuclei, and regions.

**Olfactory bulb**

| | |
|---|---|
| AOB | accessory olfactory bulb |
| AOD | anterior olfactory nucleus, dorsal part |
| AOE | anterior olfactory nucleus, external part |
| AOL | anterior olfactory nucleus, lateral part |
| AOM | anterior olfactory nucleus, medial part |
| AOV | anterior olfactory nucleus, ventral part |
| EPl | external plexiform layer of the olfactory bulb |
| EPlA | external plexiform layer of the accessory olfactory bulb |
| GlA | glomerular layer of the accessory olfactory bulb |
| GrA | granule cell layer of the accessory olfactory bulb |
| GrO | granular cell layer of the olfactory bulb |
| lo | lateral olfactory tract |
| LOT | nucleus of the lateral olfactory tract |
| Mi | mitral cell layer of the olfactory bulb |
| MiA | mitral cell layer of the accessory olfactory bulb |
| ON | olfactory nerve layer |
| vn | vomeronasal nerve |

**Cerebral cortex**

| | |
|---|---|
| AIP | agranular insular cortex, posterior part |
| Au1 | primary auditory cortex |
| AuD | secondary auditory cortex, dorsal area |
| AuV | secondary auditory cortex, ventral area |
| Cl | claustrum |
| Cg/RS | cingular/retrosplenial cortex |
| DEn | dorsal endopiriform nucleus |
| DI | dysgranular insular cortex |
| DLO | dorsolateral orbital cortex |
| Ect | ectorhinal cortex |
| FrA | frontal association cortex |
| GI | granular insular cortex |
| LEnt | lateral entorhinal cortex |
| LO | lateral orbital cortex |
| LPtA | lateral parietal association cortex |
| M1 | primary motor cortex |
| M2 | secondary motor cortex |
| MEnt | medial entorhinal cortex |

*Continued on next page*

*Continued*

**Cerebral cortex**

| | |
|---|---|
| MO | medial orbital cortex |
| MPtA | medial parietal association cortex |
| Pir | piriform cortex |
| PRh | perirhinal cortex |
| PrL | prelimbic cortex |
| RSA | retrosplenial agranular cortex |
| RSG | retrosplenial granular cortex |
| S1 | primary somatosensory cortex |
| S1BF | primary somatosensory cortex, barrel field |
| S1DZ | primary somatosensory cortex, dysgranular region |
| S1FL | primary somatosensory cortex, forelimb region |
| S1HL | primary somatosensory cortex, hindlimb region |
| S1ShNc | primary somatosensory cortex, shoulder/neck region |
| S1Tr | primary somatosensory cortex, trunk region |
| S2 | secondary somatosensory cortex |
| TeA | temporal association cortex |
| V1 | primary visual cortex |
| V2L | secondary visual cortex, lateral area |
| V2ML | secondary visual cortex, mediolateral area |
| V2MM | secondary visual cortex, mediomedial area |
| VEn | ventral endopiriform nucleus |
| VO | ventral orbital cortex |

**Forebrain**

| | |
|---|---|
| AC | anterior commissural nucleus |
| aci | anterior commissure, intrabulbar part |
| BSTIA | bed nucleus of the stria terminalis, intraamygdaloid division |
| BSTMPL | bed nucleus of the stria terminalis, medial division, posterolateral part |
| BSTMV | bed nucleus of the stria terminalis, medial division, ventral part |
| BSTS | bed nucleus of stria terminalis, supracapsular part |
| IPAC | interstitial nucleus of the posterior limb of the anterior commissure |
| LPO | lateral preoptic area |
| LSV | lateral septal nucleus, ventral part |
| MCPO | magnocellular preoptic nucleus |
| MPA | medial preoptic area |
| MPOC | medial preoptic nucleus, central part |
| MPOM | medial preoptic nucleus, medial part |
| SFO | subfornical organ |
| st | stria terminalis |
| VLPO | ventrolateral preoptic nucleus |

## Hippocampus

| | |
|---|---|
| CA1 | field ca1 of hippocampus |
| CA2 | field ca2 of hippocampus |
| CA3 | field CA3 of hippocampus |
| DG | dentate gyrus |
| f | fornix |
| fi | fimbria of the hippocampus |
| GrDG | granular layer of the dentate gyrus |
| LMol | lacunosum moleculare layer of the hippocampus |
| Mol | molecular layer of the dentate gyrus |
| Or | oriens layer of the hippocampus |
| PoDG | polymorph layer of the dentate gyrus |
| Py | pyramidal tract |
| S | subiculum |
| SLu | stratum lucidum, hippocampus |
| vhc | ventral hippocampal commissure |

## Thalamus

| | |
|---|---|
| AD | anterodorsal thalamic nucleus |
| AM | anteromedial thalamic nucleus |
| AMV | anteromedial thalamic nucleus, ventral part |
| Ang | angular thalamic nucleus |
| APTD | anterior pretectal nucleus, dorsal part |
| APTV | anterior pretectal nucleus, ventral part |
| AVDM | anteroventral thalamic nucleus, dorsomedial part |
| AVVL | anteroventral thalamic nucleus, ventrolateral part |
| CL | centrolateral thalamic nucleus |
| CM | central medial thalamic nucleus |
| DLG | dorsal lateral geniculate nucleus |
| eml | external medullary lamina |
| Eth | ethmoid thalamic nucleus |
| F | nucleus of the fields of Forel |
| fr | fasciculus retroflexus |
| IAD | interanterodorsal thalamic nucleus |
| IAM | interanteromedial thalamic nucleus |
| ic | internal capsule |
| IGL | intergeniculate leaf |
| IMA | intramedullary thalamic area |
| IMD | intermediodorsal thalamic nucleus |
| LDDM | laterodorsal thalamic nucleus, dorsomedial part |
| LDVL | laterodorsal thalamic nucleus, ventrolateral part |

*Continued on next page*

*Continued*

**Thalamus**

| | |
|---|---|
| LGP | lateral globus pallidus |
| LHb | lateral habenular nucleus |
| LPLR | lateral posterior thalamic nucleus, laterorostral part |
| LPMC | lateral posterior thalamic nucleus, mediocaudal part |
| LPMR | lateral posterior thalamic nucleus, mediorostral part |
| MD | mediodorsal thalamic nucleus |
| MDC | mediodorsal thalamic nucleus, central part |
| MDL | mediodorsal thalamic nucleus, lateral part |
| MDM | mediodorsal thalamic nucleus, medial part |
| MGP | medial globus pallidus (entopeduncular nucleus) |
| MHb | medial habenular nucleus |
| MPT | medial pretectal nucleus |
| OPT | olivary pretectal nucleus |
| PC | paracentral thalamic nucleus |
| PF | parafascicular thalamic nucleus |
| Po | posterior thalamic nuclear group |
| PPT | posterior pretectal nucleus |
| PR | prerubral field |
| PrC | precommissural nucleus |
| PT | paratenial thalamic nucleus |
| pv | periventricular fiber system |
| PV | paraventricular thalamic nucleus |
| PVA | paraventricular thalamic nucleus, anterior part |
| PVP | paraventricular thalamic nucleus, posterior part |
| Re | reuniens thalamic nucleus |
| REth | retroethmoid nucleus |
| Rh | rhomboid thalamic nucleus |
| RI | rostral interstitial nucleus of medial longitudinal fasciculus |
| Rt | reticular thalamic nucleus |
| Sc | scaphoid thalamic nucleus |
| sm | stria medullaris of the thalamus |
| SPF | subparafascicular thalamic nucleus |
| SPFPC | subparafascicular thalamic nucleus, parvicellular part |
| STh | subthalamic nucleus |
| str | superior thalamic radiation |
| Sub | submedius thalamic nucleus |
| SubG | subgeniculate nucleus |
| VA | ventral anterior thalamic nucleus |

*Continued on next page*

*Continued*

**Thalamus**

| | |
|---|---|
| VL | ventrolateral thalamic nucleus |
| VLG | ventral lateral geniculate nucleus |
| VLGMC | ventral lateral geniculate nucleus, magnocellular part |
| VLGPC | ventral lateral geniculate nucleus, parvicellular part |
| VM | ventromedial thalamic nucleus |
| VPL | ventral posterolateral thalamic nucleus |
| VPM | ventral posteromedial thalamic nucleus |
| VRe | ventral reuniens thalamic nucleus |
| Xi | xiphoid thalamic nucleus |

**Hypothalamus**

| | |
|---|---|
| AAD | anterior amygdaloid area, dorsal part |
| ACo | anterior cortical amygdaloid nucleus |
| AHA | anterior hypothalamic area, anterior part |
| AHC | anterior hypothalamic area, central part |
| AHiAL | amygdalohippocampal area, anterolateral part |
| AHiPM | amygdalohippocampal area, posteromedial part |
| AHP | anterior hypothalamic area, posterior part |
| APir | amygdalopiriform transition area |
| Arc | arcuate hypothalamic nucleus |
| ArcMP | arcuate hypothalamic nucleus, medial posterior part |
| AStr | amygdalostriatal transition area |
| BLA | basolateral amygdaloid nucleus, anterior part |
| BLP | basolateral amygdaloid nucleus, posterior part |
| BLV | basolateral amygdaloid nucleus, ventral part |
| BMA | basolateral amygdaloid nucleus, anterior part |
| BMP | basomedial amygdaloid nucleus, posterior part |
| CeC | central amygdaloid nucleus, capsular part |
| CeL | central amygdaloid nucleus, lateral division |
| CeM | central amygdaloid nucleus, medial division |
| CeMPV | central amygdaloid nucleus, medial posteroventral part |
| cp | cerebral peduncle, basal part |
| CxA | cortex-amygdala transition zone |
| DM | dorsomedial hypothalamic nucleus |
| FF | fields of Forel |
| I | intercalated nuclei of the amygdala |
| IM | intercalated amygdaloid nucleus, main part |
| LA | lateroanterior hypothalamic nucleus |
| La | lateral amygdaloid nucleus |

*Continued on next page*

*Continued*

**Hypothalamus**

| | |
|---|---|
| LaDL | lateral amygdaloid nucleus, dorsolateral part |
| LaVL | lateral amygdaloid nucleus, ventrolateral part |
| LaVM | lateral amygdaloid nucleus, ventromedial part |
| LH | lateral hypothalamic area |
| LM | lateral mammillary nucleus |
| MCLH | magnocellular nucleus of the lateral hypothalamus |
| ME | median eminence |
| MeA | medial amygdaloid nucleus, anterior part |
| MeAD | medial amygdaloid nucleus, anteriodorsal part |
| MePD | medial amygdaloid nucleus, posterodorsal part |
| MePV | medial amygdaloid nucleus, posteroventral part |
| ML | medial mammillary nucleus, lateral part |
| MM | medial mammillary nucleus, medial part |
| MMn | medial mammillary nucleus, median part |
| mt | mammillothalamic tract |
| mtg | mammillotegmental tract |
| ns | nigrostriatal bundle |
| opt | optic tract |
| PaAP | paraventricular hypothalamic nucleus, anterior parvicellular part |
| PaDC | paraventricular hypothalamic nucleus, dorsal cap |
| PaLM | paraventricular hypothalamic nucleus, lateral magnocellular part |
| PaMM | paraventricular hypothalamic nucleus, medial magnocellular part |
| PaMP | paraventricular hypothalamic nucleus, medial parvicellular part |
| PaPo | paraventricular hypothalamic nucleus, posterior part |
| PH | posterior hypothalamic area |
| PLCo | posterolateral cortical amygdaloid nucleus |
| pm | principal mammillary tract |
| PMCo | posteromedial cortical amygdaloid nucleus (C3) |
| PMD | premammillary nucleus, dorsal part |
| PR | prerubral field |
| PSTh | parasubthalamic nucleus |
| SCh | suprachiasmatic nucleus |
| SLEAC | sublenticular extended amygdala, central part |
| SLEAM | sublenticular extended amygdala, medial part |
| SO | supraoptic nucleus |
| SOR | supraoptic nucleus, retrochiasmatic part |
| SPa | subparaventricular zone of the hypothalamus |
| SubI | subincertal nucleus |

*Continued*

**Hypothalamus**

| | |
|---|---|
| SuML | supramammillary nucleus, lateral part |
| SuMM | supramammillary nucleus, medial part |
| sumx | supramammillary decussation |
| TC | tuber cinereum area |
| VMHC | ventromedial hypothalamic nucleus, central part |
| VMHDM | ventromedial hypothalamic nucleus, dorsomedial part |
| VMHVL | ventromedial hypothalamic nucleus, ventrolateral part |
| VTM | ventral tuberomammillary nucleus |
| ZI | zona incerta |
| ZID | zona incerta, dorsal part |
| ZIV | zona incerta, ventral part |

**Midbrain and pons**

| | |
|---|---|
| Acs5 | accessory trigeminal nucleus |
| CIC | central nucleus of the inferior colliculus |
| CnF | cuneiform nucleus |
| DCIC | dorsal cortex of the inferior colliculus |
| DMPAG | dorsomedial periaqueductal gray |
| DMTg | dorsomedial tegmental area |
| DpMe | deep mesencephalic nucleus |
| DRC | dorsal raphe nucleus, caudal part |
| DRD | dorsal raphe nucleus, dorsal part |
| DRI | dorsal raphe nucleus, interfascicular part |
| DRV | dorsal raphe nucleus, ventral part |
| DTgC | dorsal tegmental nucleus, central part |
| DTgP | dorsal tegmental nucleus, pericentral part |
| ECIC | external cortex of the inferior colliculus |
| EMi | epimicrocellular nucleus |
| I5 | intertrigeminal nucleus |
| InCo | intercollicular nucleus |
| IPF | interpeduncular fossa |
| IPI | interpeduncular nucleus, intermediate subnucleus |
| KF | Ko¨lliker-Fuse nucleus |
| LDTg | laterodorsal tegmental nucleus |
| LDTgV | laterodorsal tegmental nucleus, ventral part |
| LPAG | lateral periaqueductal gray |
| LPBC | lateral parabrachial nucleus, central part |
| LPBD | lateral parabrachial nucleus, dorsal part |
| LPBE | lateral parabrachial nucleus, external part |

*Continued on next page*

*Continued*

**Midbrain and pons**

| | |
|---|---|
| LPBS | lateral parabrachial nucleus, superior part |
| LPBV | lateral parabrachial nucleus, ventral part |
| MCPC | magnocellular nucleus of the posterior commissure |
| Me5 | mesencephalic trigeminal nucleus |
| MGV | medial geniculate nucleus, ventral part |
| MiTg | microcellular tegmental nucleus |
| ml | medial lemniscus |
| mlf | medial longitudinal fasciculus |
| MnR | median raphe nucleus |
| Mo5 | motor trigeminal nucleus |
| MPB | medial parabrachial nucleus |
| P5 | peritrigeminal zone |
| PAG | periaqueductal gray |
| PC5 | parvicellular motor trigeminal nucleus |
| pc | posterior commissure |
| PCom | nucleus of the posterior commissure |
| PMnR | paramedian raphe nucleus |
| PnC | pontine reticular nucleus, caudal part |
| PnO | pontine reticular nucleus, oral part |
| PnR | pontine raphe nucleus |
| PnV | pontine reticular nucleus, ventral part |
| PPTg | pedunculopontine tegmental nucleus |
| RC | raphe cap |
| RPF | retroparafascicular nucleus |
| RtTg | reticulotegmental nucleus of the pons |
| RtTgP | reticulotegmental nucleus of the pons, pericentral part |
| Sag | sagulum nucleus |
| scp | superior cerebellar peduncle (brachium conjunctivum) |
| SNC | substantia nigra, compact part |
| SNR | substantia nigra, reticular part |
| SPTg | subpedencular tegmental nucleus |
| Su5 | supratrigeminal nucleus |
| SubCD | subcoeruleus nucleus, dorsal part |
| SubCV | subcoeruleus nucleus, ventral part |
| ts | tectospinal tract |
| Tz | nucleus of the trapezoid body |
| VLPAG | ventrolateral periaqueductal gray |
| VTA | ventral tegmental area |
| VTg | ventral tegmental nucleus |

**Midbrain and pons**

| | |
|---|---|
| xscp | decussation of the superior cerebellar peduncle |

**Medulla**

| | |
|---|---|
| 7N | facial nucleus |
| 10N | dorsal motor nucleus of vagus |
| 12N | hypoglossal nucleus |
| Amb | ambiguus nucleus |
| AP | area postrema |
| Cu | cuneate nucleus |
| DLL | dorsal nucleus of the lateral lemniscus |
| DMSp5 | dorsomedial spinal trigeminal nucleus |
| DPO | dorsal periolivary region |
| ECu | external cuneate nucleus |
| Gi | gigantocellular reticular nucleus |
| icp | inferior cerebellar peduncle (restiform body) |
| ILL | intermediate nucleus of the lateral lemniscus |
| In | intercalated nucleus of the medulla |
| IOBe | inferior olive, subnucleus B of medial nucleus |
| IOC | inferior olive, subnucleus C of medial nucleus |
| IOD | inferior olive, dorsal nucleus |
| IOK | inferior olive, cap of Kooy of the medial nucleus |
| IRt | intermediate reticular nucleus |
| lfp | longitudinal fasciculus of the pons |
| LPGi | lateral paragigantocellular nucleus |
| LRt | lateral reticular nucleus |
| LSO | lateral superior olive |
| LVPO | lateroventral periolivary nucleus |
| MdD | medullary reticular nucleus, dorsal part |
| MdV | medullary reticular nucleus, ventral part |
| ml | medial lemniscus |
| MVe | medial vestibular nucleus |
| MVPO | medioventral periolivary nucleus |
| PCRt | parvicellular reticular nucleus |
| PL | paralemniscal nucleus |
| PMn | paramedian reticular nucleus |
| Pr | prepositus nucleus |
| Pr5 | principal sensory trigeminal nucleus |
| Pr5DM | principal sensory trigeminal nucleus, dorsomedial part |
| Pr5VL | principal sensory trigeminal nucleus, ventrolateral part |
| PSol | parasolitary nucleus |

*Continued on next page*

*Continued*

**Medulla**

| | |
|---|---|
| py | pyramidal tract |
| RMg | raphe magnus nucleus |
| RPa | raphe pallidus nucleus |
| RPO | rostral periolivary region |
| RVL | rostroventrolateral reticular nucleus |
| s5 | sensory root of the trigeminal nerve |
| SolC | nucleus of the solitary tract, commissural part |
| SolCe | nucleus of the solitary tract, central part |
| SolDL | solitary nucleus, dorsolateral part |
| SolDM | nucleus of the solitary tract, dorsomedial part |
| SolG | nucleus of the solitary tract, gelatinous part |
| SolI | nucleus of the solitary tract, interstitial part |
| SolIM | nucleus of the solitary tract, intermediate part |
| SolM | nucleus of the solitary tract, medial part |
| SolV | solitary nucleus, ventral part |
| SolVL | nucleus of the solitary tract, ventrolateral part |
| sp5 | spinal trigeminal tract |
| Sp5I | spinal trigeminal nucleus, interpolar part |
| SPO | superior paraolivary nucleus |
| SpVe | spinal vestibular nucleus |
| VCA | ventral cochlear nucleus, anterior part |
| vsc | ventral spinocerebellar tract |

**Cerebellum**

| | |
|---|---|
| 6Cb | 6th Cerebellar lobule |
| 7Cb | 7th Cerebellar lobule |
| 9Cb | 9th Cerebellar lobule |
| Ant | anterior lobe cerebellum |
| Crus1 | crus 1 of the ansiform lobule |
| Crus2 | crus 2 of the ansiform lobule |
| Fl | flocculus |
| mcp | middle cerebellar peduncle |
| PFl | paraflocculus |
| PM | paramedian lobule |
| Sim | simple lobule |

**Ventricular zones**

| | |
|---|---|
| 3V | 3rd ventricle |
| OV | olfactory ventricle (olfactory part of lateral ventricle) |

