## [Editor Report]

This study provides a valuable, transcript-level map of OXT neurons and OXTR expression across the mammalian brain using advanced single-molecule RNAscope. The authors present compelling evidence supporting their conclusions, combining chromogenic assays with high-quality, state-of-the-art microscopy. By clearly delineating oxt and oxtr expression across multiple nuclei and brain regions relevant to behavior and physiology, the work substantially advances understanding of central oxytocin signaling and will be of broad interest to neuroscientists and endocrinologists.

---

## [Decision Letter]

**Decision letter after peer review:**

Thank you for submitting your article "Single Transcript Level Atlas of Oxytocin and the Oxytocin Receptor in the Mouse Brain" for *eLife*'s consideration. Your article has been reviewed by 2 peer reviewers, one of whom is a member of our Board of Reviewing Editors, and the evaluation has been overseen by Ma-Li Wong as the Senior Editor.

Essential Revision:

1) We are not asking for additional data, but please clearly explain Figure#1 to address Reviewer#2's comments on the nuclei and subnuclei.

2) Please also revise the text following Reviewer#2's comments, including the number of females and males.

*Reviewer #1 (Recommendations for the authors):*

This is a beautiful study with compelling evidence to reveal Oxt and Oxtr expressions in mouse brains and the differences between males and females. The information provided in this study provides a platform to define new functional roles of brain-region-specific Oxt and Oxtr in rodent behaviors, such as anxiety- and depressive-like behaviors, energy metabolism, such as obesity, and other OXT-related phenotypes in health and disease.

*Reviewer #2 (Recommendations for the authors):*

This manuscript is clearly written. Strengths of this paper include: 1) The data and figures clearly show the expression level of Oxt and Oxtr in different brain nuclei and regions; 2) Sex difference in Oxt expression is also well demonstrated; 3) Extensions to the functions of OXT in CNS regulation are sufficiently discussed. In all, this will provide the background and rationale to investigate further the role of OXT in CNS regulation on different behaviors and physiological processes. However, some missing points need to be addressed in the manuscript:

The results do not sufficiently demonstrate the Oxt expression level in the whole brain. The authors should consider making another figure showing Oxt transcript density in nuclei, sub-nuclei and regions of different brain parts, just like figure 1. If not, please provide a reasonable explanation.

In Results, from lines 155 to 158, the authors talked about the highly expressed Oxtr in LH and DM of the hypothalamus and its probable impact on bone and fat tissue through SNS outflow. Firstly, it would be better to put this part in the Discussion section. Secondly, since you have given a large piece of introduction on the probable function of OXT on the regulation of skeleton and body composition in both the Abstract and Introduction sections (from lines 85 to 94), this would be a significant part of your paper to discuss after the discovery of Oxtr expression in LH and DM. In the Discussion, it is necessary to elaborate more on its probable association with bone and fat regulation. More literature should be added, such as OXT's function on leanness, which can be related to food intake and ingestive behaviors you mentioned in the Discussion.

In the last paragraph of Results, the authors mentioned the overlapping expression of Oxt and Oxtr in some regions. Presenting representative RNAscope images of these regions (PaMP, PaMM, PaLM, Arc, MeA, TC, and LA) would be more convincing.

In Methods, please mention the number of female and male mouse brains used in the study.

---

## [Author Response]

Essential Revision:Reviewer #1 (Recommendations for the authors):This is a beautiful study with compelling evidence to reveal Oxt and Oxtr expressions in mouse brains and the differences between males and females. The information provided in this study provides a platform to define new functional roles of brain-region-specific Oxt and Oxtr in rodent behaviors, such as anxiety- and depressive-like behaviors, energy metabolism, such as obesity, and other OXT-related phenotypes in health and disease.

We thank the reviewer for the kind words.

Reviewer #2 (Recommendations for the authors):This manuscript is clearly written. Strengths of this paper include: 1) The data and figures clearly show the expression level of Oxt and Oxtr in different brain nuclei and regions; 2) Sex difference in Oxt expression is also well demonstrated; 3) Extensions to the functions of OXT in CNS regulation are sufficiently discussed. In all, this will provide the background and rationale to investigate further the role of OXT in CNS regulation on different behaviors and physiological processes. However, some missing points need to be addressed in the manuscript:The results do not sufficiently demonstrate the Oxt expression level in the whole brain. The authors should consider making another figure showing Oxt transcript density in nuclei, sub-nuclei and regions of different brain parts, just like figure 1. If not, please provide a reasonable explanation.

We thank the reviewer for the comment. Unlike the oxytocin receptor that is widely expression in the brain, we find that the expression of oxytocin is limited to certain sub-regions of the hypothalamus and the forebrain express *Oxt*. What we present in the manuscript is the complete expression profile of *Oxt*. Nonetheless, to make this study more comprehensive, we have now included the expression of *Oxtr* in female mice in Figure 2 and Supplementary Figure 2.

In Results, from lines 155 to 158, the authors talked about the highly expressed Oxtr in LH and DM of the hypothalamus and its probable impact on bone and fat tissue through SNS outflow. Firstly, it would be better to put this part in the Discussion section. Secondly, since you have given a large piece of introduction on the probable function of OXT on the regulation of skeleton and body composition in both the Abstract and Introduction sections (from lines 85 to 94), this would be a significant part of your paper to discuss after the discovery of Oxtr expression in LH and DM.

As suggested, we have moved this part from the Results to the Discussion section.

In the Discussion, it is necessary to elaborate more on its probable association with bone and fat regulation. More literature should be added, such as OXT's function on leanness, which can be related to food intake and ingestive behaviors you mentioned in the Discussion.

We have now added two paragraphs on the association of oxytocin with bone and fat regulation (please see pages 10–11, lines 196–204 and pages 14–15, lines 290–306).

In the last paragraph of Results, the authors mentioned the overlapping expression of Oxt and Oxtr in some regions. Presenting representative RNAscope images of these regions (PaMP, PaMM, PaLM, Arc, MeA, TC, and LA) would be more convincing.

We thank the reviewer for this comment. RNAscope for *Oxt* and *Oxtr* has been performed separately, therefore, regretfully, we do not have images showing colocalization of *Oxt* and *Oxtr*.

In Methods, please mention the number of female and male mouse brains used in the study.

The number of mice has now been stated in the Results (page 6, line 103) and Materials and methods (page 16, line 324) sections.